# Physical and Mechanical Behavior of New Ternary and Hybrid Eco-Cements Made from Construction and Demolition Waste

**DOI:** 10.3390/ma16083093

**Published:** 2023-04-14

**Authors:** Moisés Frías, Manuel Monasterio, Jaime Moreno-Juez

**Affiliations:** 1Eduardo Torroja Institute for Construction Sciences (IETcc), Spanish National Research Council (CSIC), 28033 Madrid, Spain; 2TECNALIA, Basque Research and Technology Alliance (BRTA), Astondo Bidea, Edificio 700, Parque Tecnológico de Bizkaia, 48160 Derio, Spain

**Keywords:** CDWs, ternary eco-cements, CSA eco-cements, synergy of hybrid cements, characterization, physical properties, mechanical behaviour, microporosity

## Abstract

Construction and demolition waste (CDW) currently constitutes a waste stream with growing potential use as a secondary raw material in the manufacture of eco-cements that offer smaller carbon footprints and less clinker content than conventional cements. This study analyzes the physical and mechanical properties of two different cement types, ordinary Portland cement (OPC) and calcium sulfoaluminate (CSA) cement, and the synergy between them. These cements are manufactured with different types of CDW (fine fractions of concrete, glass and gypsum) and are intended for new technological applications in the construction sector. This paper addresses the chemical, physical, and mineralogical characterization of the starting materials, as well as the physical (water demand, setting time, soundness, water absorption by capillary action, heat of hydration, and microporosity) and mechanical behavior of the 11 cements selected, including the two reference cements (OPC and commercial CSA). From the analyses obtained, it should be noted that the addition of CDW to the cement matrix does not modify the amount of water by capillarity with respect to OPC cement, except for Labo CSA cement which increases by 15.7%, the calorimetric behavior of the mortars is different depending on the type of ternary and hybrid cement, and the mechanical resistance of the analysed mortars decreases. The results obtained show the favorable behavior of the ternary and hybrid cements made with this CDW. Despite the variations observed in the different types of cement, they all comply with the current standards applicable to commercial cements and open up a new opportunity to improve sustainability in the construction sector.

## 1. Introduction

Recent years have brought growing interest in the use of industrial waste as a secondary raw material in the manufacture of new, more sustainable, and more environmentally friendly eco-cements [1,2,3,4]. This trend is driven by recent strategies relating to the circular economy, the Green Deal 2030, climate neutrality, and the 5 Cs roadmap for the cement industry, among others [5,6,7,8]. One of the waste streams of greatest interest to the construction sector is construction and demolition waste (CDW) [9,10,11], because of the enormous annual volume generated (around 350–400 million tons in Europe) that, in most cases, is disposed of in landfill and engenders a series of environmental, technical, economic and health consequences [12]. Concrete and mixed CDW currently has an industrial application as a recycled aggregate (fine or coarse), as endorsed by a variety of international regulations [13].

In 2021, European standard EN 197-6 [14] introduced a new use for the fine fraction of recycled concrete as an alternative pozzolana in the manufacture of commercial type-II eco-cements containing up to 20% replacement content. Previous studies have demonstrated the scientific and technical feasibility of recycling the fine fractions of concrete CDW (<5 mm), or even recycled glass mixtures, for use as active additions [15,16,17,18,19]. Their use as replacements for natural materials (bauxite and gypsum) in the manufacture of calcium sulfoaluminate (CSA) eco-cements has also been analyzed [20]. All these cements have been eco-designed considering environmental criteria and guaranteeing the reduction of their carbon footprint. The analysis of the life cycle of the eco-cements and the calculation of the embodied carbon footprint reduction are carried out in other works carried out by the authors, concluding that, for the ternary cements, the overall environmental impact could be reduced in the same proportion as the replacement rate [17,21].

In light of these findings, research has focused on investigating new cements with a low carbon footprint and on applying them in new construction technologies, a field that, from a scientific and technical standpoint, has yet to be extensively explored. This paper analyzes for the first time the physical and mechanical behavior of these new eco-cements and the synergy between the various hybrid cements, ordinary Portland cement (OPC), and CSA, all made from CDW. These cements are then compared with the respective commercial cements.

## 2. Materials and Methods

### 2.1. Raw Materials

The different types of eco-cements studied in this work were manufactured by blending different types of CDW and cements. The starting materials used to obtain the different types of eco-cements were as follows:

Four CDW materials: (i) Two fine fractions (<5 mm) of concrete waste selected to be representative of what is generated in the central and northern regions of Spain: a siliceous concrete (HsT) waste provided by the company TECREC (Madrid, Spain) and a calcareous concrete waste (HcG) provided by the company GUTRAM (Basque Country, Spain). Both fine waste materials were obtained from the valorization process (crushing of the input fraction for the obtention of coarse recycled aggregates 4–20 mm and screening of the fine fraction <4 mm) of CDW corresponding to the European Waste Code (EWC) “170101-Concrete”. These fine fractions are rejections of the process and are usually stored at CDW management plants without any viable industrial application, (ii) A 100% amorphous recycled laminated glass obtained from the demolition of a residential building representative of the most common residual glass fraction obtained in demolitions. (iii) a recycled gypsum (plaster) with a particle size below 200 μm with a purity higher than 99%. Further information on these CDW materials is found in previous papers [15,16]. All the waste used in this research paper was dried and ground to below 63 μm at laboratory to obtain a similar granulometry to OPC.

Two commercial cements: An OPC cement (type CEM I 52.5 R) supplied by the company Cementos Lemona, S.A. (Bilbao, Spain) and a commercial CSA cement (Come CSA) sold under the brand “i.tech ALI PRE GREEN” by Heidelberg Cement Group (Heidelberg, Germany) [17].

One laboratory-scale CSA cement: A calcium sulfoaluminate (CSA) cement clinkerized in an electric muffle furnace in a laboratory at 1250 °C for 60 min (Labo CSA) using 20% CDW (composed of HsT, HcG, and glass) as a replacement for bauxite. Within the 20% replacement content the ternary admixture consists of HsT, HcG, and glass at a 80%/10%/10% ratio by weight, respectively. Finally, the Labo CSA cement was clinkerized using 28.5% recycled gypsum (laminated plasterboard) as a source of sulfate to replace natural gypsum. The rest of the components were high-quality commercial laboratory products. The detailed process for the pre-clinkerization phase is found in [20].

Table 1 shows the chemical compositions obtained for all the materials using X-ray fluorescence (XRF), emphasizing the differences in chemical composition according to the nature of the material.

The mineralogical compositions of the two CSA cements identified using the X-ray diffraction (XRD)/Rietveld method are shown in Table 2, revealing the difference in that the CSA cement clinkerized with CDW contains high levels of brediggite (27.9%) and gehlenite (30.2%) content. The ye’elemite/C_2_S ratios are 29.15 and 1.48 for the Come CSA and Labo CSA cements, respectively.

The mineralogical analysis of the starting materials, using XRD, are shown in Figure 1. In the concrete waste materials, quartz is the predominant mineral identified in the HsT waste, while calcite is the main mineral identified in the HcG waste, this content being a consequence of the different natures of the initially used natural aggregates. Finally, this mineralogical phase is clearly evident in the recycled gypsum sample’s diffraction peaks at 11.5, 21, and 29-2θ. The recycled glass, meanwhile, has a totally amorphous nature. The rutile’s diffraction peaks are located at 27.5, 36.5, 39, 54, 56, 62.5, and 64-2θ due to its use as the internal standard. In the two CSA cements (Come and Labo), the main mineralogical phases identified are ye’elemite, located at 24, 27.5, 30, 33, and 42-2θ, anhydrite, located at 28, 31, and 57-2θ, and belite, at 32.5-2θ. Although not shown in the figure, the Labo CSA cement also contains brediggite (32.80, 33.68, and 39.82-2θ) and gehlenite (31.36-2θ).

As regards the granulometry identified by laser diffraction, Table 3 shows the Dx(10), Dx(50), and Dx(90) values for all the materials except the OPC and Come CSA commercial cements after grinding and screening at 63 μm.

### 2.2. Ternary and Hybrid Cement Preparation

The ternary cements were prepared by replacing the reference OPC cement with 7% of each of the binary pozzolanic admixtures in a 2:1 ratio of concrete/glass waste (7% HsT and 7% HcG) as per the prior optimization analysis [18,19].

The hybrid cements were made by replacing the ternary cements with 10% of each of the CSA cements (7% HsT/HcG + Come/Labo CSA). In addition to these cement mixtures used to perform a comparative analysis, other cement mixtures were also used in which the reference OPC cement was replaced by 10% CSA cement (Come and Labo) and pure CSA cement (Labo CSA and Come CSA) with the aim of assessing the influence of each of the cement components on the hybrid eco-cements. The mixtures of the 11 cements selected for this paper are shown in Table 4.

### 2.3. Characterization Performed on Eco-Cements

#### 2.3.1. Physical and Mechanical Properties

The physical tests performed on the cement pastes (water required for normal consistency, initial setting, and expansion) and the mechanical tests performed on the cement mortars (compressive strength) were conducted in accordance with currently applicable European standards (EN 196-1 and EN 196-3) [22,23]. The mortar specimens were tested in an IBERTEST AUTOTEST 200/20-SW press (Ibertest, Sapin) at 2 and 28 days of curing in water.

##### Water Absorption Due to Capillary Action

The mortars’ capillary water absorption capacity was analyzed as per the Fagerlund method, as described in Spanish standard UNE 83982 [24], using specimens measuring 4 × 4 × 16 cm and cured for 28 days. The specimens were then pre-conditioned in several stages as per Spanish standard UNE 83,966 [25] to obtain homogenous moisture distribution throughout them (65–75%). After this pre-conditioning, the specimens were placed in a container and partially immersed in water (up to a height of 5 mm). The weight gain was then measured at different test intervals. This weight gain was used to calculate the sorptivity coefficient (*S*), achieved via Equation (1) [26].
(1)$$\frac{W}{A}=S_{0}+S\sqrt[{2}]{t}$$
where *W* represents the amount of water absorbed, *A* is the area of the specimen exposed to water, and *t* is time. *S*_0_ is used as a correction coefficient for the initial amount of water absorbed by the pores. Another more recent model by which to represent capillary water absorption is the one proposed by Villagrán et al. [27]. This model establishes a linear relationship between the results using the fourth root of time, as opposed to the previous model, which uses the square root. This Villagrán model uses Equation (2):(2)$$\frac{W}{A}=S\cdot \sqrt[{4}]{t}$$

#### 2.3.2. Langavant Semi-Adiabatic Calorimetry

The heat of hydration was determined using the semi-adiabatic method specified in European standard EN 196-9 [28]. The heat produced in hydrating the mortar is collected and compared with a reference inert mortar mixed 12 months previously. The value obtained is then entered in Equation (3) to determine the heat of hydration (*Q*):(3)$$Q=\frac{C}{m_{c}}\theta_{t}+\frac{1}{m_{c}}\int_{0}^{t}\alpha \cdot \theta_{t}\cdot dt$$
where *Q* is the heat of hydration in Jg^−1^, *m_c_* is the mass of the mortar for which the heat of hydration is to be calculated in *g*, *t* is hydration time in hours, *C* is the total heat capacity of the calorimeter and the sample mortar in J°C, *α* is the global heat transmission coefficient in Jh^−1^ °C^−1^, and *θ_t_* is the difference between the calorimeter readings for the sample mortar and the control piece at time *t*.

### 2.4. Instrumental Techniques

An XRF spectrometer, model Philips PW-1404 (Phillips, Madrid, Spain), equipped with an Sc-Mo X-ray tube, was used to determine the chemical composition of the starting materials and the blended cements.

Particle size distribution was performed by laser diffraction on a Malvern Mastersizer 3000 analyzer (Malvern Panalytical, Madrid, Spain) equipped with red and blue light sources (He-Ne and LED) in dry dispersion mode. The measurement range was from 0.01 to 3500 µm.

Heating and heat of hydration were quantified by applying the Langavant semi-adiabatic procedure as per European standard recommendations [28]. An inert (more than 12 months old) reference cement was placed in the inner bottle of a Dewar vacuum flask and the paste to be tested was placed in the outer one. Heating, defined as the difference in temperature between the two, was used to calculate the heat of hydration. The calorimeter model employed was an Ibertest IB32-101E equipped with WinLect32 software.

The mineralogical analyses were carried out by powder XRD on a PAN Analytical X’Pert Pro X-ray diffractometer (Malvern Panalytical, Davis, CA, USA) fitted with a Cu anode, operating at 40 mA, 45 kV, and using a divergence slit of 0.5° with 0.5 mm reception slits. The samples were scanned in a 2θ range of 5° to 60°, with a step size of 0.0167 (2θ) at 150 ms/step. Rutile was used as the internal standard. Rietveld quantification was performed with Match v.3 and FullProf suite software (Crystal Impact, Bonn, Germany) and the mineralogical phases were identified using the Crystallography Open Database (COD).

The porosity of the pastes was analyzed using mercury intrusion porosimetry in a Micromeritics Autopore IV porosimeter (Micromeritics, Norcross, GA, USA). This device operates at pressures that reach 33,000 psi (227.5 MPa), measuring pore diameters between 0.006 and 175 µm.

## 3. Results and Discussion

### 3.1. Physical Behavior of the Blended Cement Pastes

To ascertain the influence of the CDW on the various types of eco-cement selected, the physical behavior of the cement pastes in relation to the water required for normal consistency (WD), the initial setting time (IST), and soundness (S) was analyzed in accordance with the requirements described in standard EN 197-1 [29]. Table 5 shows the results obtained for the different cements analyzed, including the minimum requirements set out in the currently applicable European standard.

The table shows that the cement pastes containing 7% CDW require less water (lower WD) than the reference OPC paste. A fluidizing effect is seen in both cases and is most pronounced when lime concrete waste (HcG) is employed. For the hybrid cements, the replacement of ternary cement (7% HsT/HcG) with 10% CSA does not instigate a change in water demand, which remains below that of the reference OPC paste, including the pure CSA cement (Come CSA). The greatest change is seen in the Labo CSA cement paste, which requires 15.7% more water to achieve a consistency similar to the one obtained by the OPC.

As regards the initial setting time values, the cement pastes exhibit different behaviors: the ternary cements and the hybrid cements made with clinkerized CSA in the laboratory (7% HsT/HcG + Labo) behave similarly to the reference OPC paste, taking into account the test error (±10 min), despite the fact that the Labo CSA cement reduces the initial setting time by around 100 min. However, the hybrid pastes made with 10% commercial CSA (7% HsT/HcG + Come) exhibit a rapid accelerator effect at the start of setting, reducing the times to 50–60 min. The values obtained exceed the minimum required by the standard for commercial cements in the 52.5 strength category (≥45 min). This notable reduction in setting time is conditioned by the ye’elimite content present in the CSA cements [30]. The ye’elimite, the main phase in this type of commercial cement, reacts quickly in the presence of gypsum and lime to form ettringite, accelerating setting and reducing it to as little as 17 min in the case of the pure Come CSA cement. As shown in Table 2, the laboratory CSA has less ye’elemite than the commercial CSA and therefore has no influence on the setting time but the commercial CSA does.

As regards the expansion values, all the ternary and hybrid cements behave similarly to the reference OPC cement, thereby complying with the currently applicable standards (≤10 mm) and concluding that the additions made have no effect on this property.

### 3.2. Water Absorption by Capillary Action

The changes in the capillary absorption capacity of the selected mortars relative to the square root of the time are shown in Figure 2.

The results show linear behavior for the first 6 h of testing, with the behavior later stabilizing after 4 days. This trend is consistent with the ASTM C1585 [31] standard, which indicates that capillary absorption comprises two phases, known as primary and secondary absorption, which occur between the first measurement and the measurement taken at 6 h, and then between the first and seventh days, respectively. The results obtained from these coefficients, applying Equation (1), are shown in Table 6.

The linear regression performed for the first absorption shows R^2^ values above 0.95 in most of the analyzed mortars. However, in some of the cases, such as the Labo CSA mortar, the R^2^ value decreases to 0.737, showing a non-linear progression of the absorption values.

As regards primary absorption, the coefficients vary widely, ranging between 0.05 and 0.97 × 10^−2^ cm/min^0.5^, without exhibiting a clear trend related to the nature of the mortars. The OPC mortars (7% HsT + Come CSA and OPC + Labo CSA) present higher absorption coefficients (approximately 0.90), while the Labo CSA, Come CSA, and 7% HsT + Labo mortars show coefficients below 0.091, values that demonstrate the minimal capillary porosity in these eco-cements.

The behavior during secondary absorption is totally different to that during primary absorption: the values obtained indicate minimal capacity to absorb water by capillary action, as the curves remained practically unchanged throughout the exposure time. This trend is reflected in the values of the coefficients that do not exceed 0.1. The exception to this trend is the Labo CSA mortar, which shows a negative value, thereby indicating that after 24 h of testing the mortar begins to lose the water absorbed by capillary action. Some authors justify the negative sorptivity values by pointing to the “noise” in the weight measurements [32]. However, in our case, the values are likely to be related to the capillary pore network itself, as the laboratory-scale clinkerized sulfoaluminate mortar (Labo CSA), as will be discussed in the section on microporosimetry, has a high percentage of macropores and a minimal percentage of capillary pores. Owing to the low capillary absorption capacity of the mortars selected in this second phase, the R^2^ values fall to between 0.7 and 0.82. These values are below the coefficients required by the ASTM C1585 standard (R^2^ > 0.98), as was the case in previous studies [33]. This problem is largely resolved by applying Villagrán’s fourth root model [27], as shown in Figure 3. Thus, sorptivity is calculated as a single absorption, obtaining more acceptable R^2^ values (Table 7), with a minimum value for the Labo CSA sample of 0.70.

The AS coefficient values according to the fourth root of the time are much higher than those obtained with the square root. In most cases, the values range between 5 and 6.5 × 10^−2^ cm/min^0.25^, coefficients that fall below 1 in the hybrid mortars (7% HsT and HcG + CSA) and in the pure CSA cements (Come and Labo). As in the previous case, no trend is observed in relation to the composition of the blended cements or the influence of the incorporated CDW. Both methodologies used to calculate the absorption coefficients agree on the fact that the mortar made with clinkerized CSA cement in the laboratory had the lowest sorptivity coefficient (0.19 × 10^−2^ cm/min^0.25^). The R^2^ results obtained in this way have substantially improved and fall mainly between 0.95 and 0.99, meaning a greater number of analyzed mortars would comply with the minimum of 0.98 set by the ASTM standard.

In general, incorporating a CDW admixture into ternary eco-cements (7% HsT/HcG) does not substantially alter the AS coefficients versus the OPC cement, although a slight increase is observed in the 7% HsT mortar. This was reported in previous studies when the CDW was incorporated in mortars as a fine aggregate [34,35,36] and in concretes as a coarse aggregate [37,38].

As regards the CSA cements, this lower capillary absorption capacity is in line with the findings of Mobili et al. [39,40], who reported lower capillary absorption figures for these cements than for the OPC mortars. In the case of the hybrid cements, capillary behavior is most similar to that of the CSA cements. With the hybrid cements containing 10% Come CSA, the differences owing to the nature of the CDW (calcareous or siliceous) are greater when the hybrid cement is made with 10% Labo CSA.

### 3.3. Heat of Hydration

The changes in the heat of hydration and the temperature inside the mortar specimens during the standardized semi-adiabatic test are shown in Figure 4.

The results show that the commercial CSA cement (Come CSA) exhibits a sharp rise in temperature and heat of hydration values for the first 7–8 h of reaction and exhibits an accelerator effect versus the reference OPC mortar, a trend that is in line with the initial setting values (Table 1). However, the CSA cement mortar clinkerized in the laboratory (Labo CSA) exhibits completely opposite calorimetric behavior owing to its greater content of less reactive phases (C_2_S, brediggite, and gehlenite).

Incorporating 10% commercial CSA cement in both the OPC cement and the hybrid cements (7% HsT/HcG + 10% Come CSA) produces very similar calorimetric behavior, revealing two different reactivities and a significant increase in the values recorded in the first 2.5 h and between 10 and 20 h. This first reactivity would be related to the ye’elimite’s high hydration reactivity to form ettringite, while the second reactivity could be related to the hydration of the anhydrous cement (93%) particles themselves and to the pozzolanic reaction of the binary pozzolanic admixtures (HsT and HcG/glass (7%)), generating less internal and hydration heat in this time range than the OPC. The other mortars exhibit similar behavior to the OPC, there being a slight calorimetric increase in the ternary cement mortars (7% HsT and HcG + glass) in the first 5 h of hydration due to the accelerator effect of the additional sodium provided by the recycled glass [41]. This slight increase in both parameters (heat of hydration and heating) is observed in the OPC + Labo CSA mortar between 5 and 15 h.

The results obtained after incorporating CDW into the manufacture of various eco-cements differ from the bibliographic results reported by Bordy et al. [42] and Medina et al. [43]. This difference could be related to the incorporation of recycled glass (Table 4), a material that because of its amorphous nature is highly reactive [44], thereby improving the pozzolanic synergy in the HsT/glass and HcG/glass binary admixtures and counterbalancing the low activity of the HsT and HcG concrete waste [18,19,45]. The results obtained for the commercial CSA cement are consistent with previous studies [46,47] and their effect is observed in the ternary mortars, as explained above. The difference in the behavior of the Come CSA and Labo CSA cements is explained by the differences in mineralogical composition set out in Table 1, which shows the different ye’elimite/C_2_S ratios and, therefore, their different rates of hydration.

### 3.4. Mechanical Properties

The compressive strength results obtained at 2 and 28 days of hydration for all the analyzed mortars are shown in Figure 5.

The incorporation of 7% CDW (concrete waste + glass) in the cement produces a decrease in the compressive strength of the ternary cement mortar in both the short term (2 d) and the medium term (28 d). No significant differences owing to the nature of the recycled concrete waste (siliceous or calcareous) were observed. This reduction in strength versus the OPC mortar ranges from 15–19% at 2 d before falling to 9–12% at 28 d, a fact that could be related, on the one hand, to the low pozzolanic activity of the concrete waste and, on the other, to the sodium provided by the recycled glass, which negatively affects the hydration and pozzolanic reaction rates [42].

The greatest mechanical differences are observed in the Labo CSA and Come CSA cements. As expected, the commercial CSA cement exhibits a 64% increase in compressive strength at 2 d versus the OPC mortar and exhibits similar values at 28 d. However, the CSA cement clinkerized in the laboratory with CDW exhibits a very significant decrease in strength at both curing ages, recording a value of 10 MPa at 28 d, well below the average for the other mortars, which stands at around 66 MPa. Paradoxically, the mechanical behavior of the hybrid mortars with 10% Labo CSA replacement content is normal, possibly due to the synergy between the two types of cement and the low proportion of replacement content (10%), which have a consequent effect on the hybrid cement’s mechanical properties. This sharp drop in the compressive strength of the Labo CSA eco-cement could be due, as mentioned above (Table 1), to the 74.01% content comprising mineralogical phases that are largely unreactive in the short and medium term (C_2_S, brediggite, gehlenite) versus the 3.17% (C_2_S) in the commercial CSA cement. One consequence of this is the cement mortar’s sandy appearance and the lack of cohesion and fluidity between its components (Figure 6).

The mechanical results obtained show that all the cement mortars made with CDW exhibit compressive strength values around 60 MPa, values above the minimum requirement set in the EN 197-1 standard (≥52.5 R). The cement would therefore be placed in the same strength category as the starting OPC cement.

### 3.5. Microporosimetry Analysis

Figure 7 shows both the total and partial porosities, obtained by mercury porosimetry, as a function of the pore interval and average pore diameter of all the mortars analyzed at 28 days of curing. The results make evident that the total porosity of the mortars containing OPC cement (ternary and hybrid) varies between 12% and 15%, showing a slight increase when the ternary cements containing CDW are added. The mortars made with the hybrid cements (7% HsT + Come/Labo and 7% HcG + Come) are mainly positioned at the upper end of the range. As regards the CSA cements, Come CSA exhibits a total porosity of 8.8%, a value well below that of the other analyzed OPC mortars. At the other end of the scale lies the Labo CSA cement obtained in the laboratory and partially synthesized with CDW, which reaches a value of 27.7%.

Analysis of the partial porosities at the different pore intervals reveals that the percentage of gel pores (<10 nm) in all the analyzed mortars, including the CSA cements, remains practically unaltered by addition of CDW, with the percentages present oscillating between 0.21% and 0.34%. As regards average capillary pore size (50–10 nm), the mortars containing OPC cement oscillate between 2.84% and 5.07%, with the mortars made with the ternary cements (7% HcG and 7% HsT) generally exhibiting the highest percentages (4.6–5.1%). As regards the hybrid cements, those based on lime concrete waste (7% HcG + Come/Labo) show slightly higher capillary porosity than their siliceous concrete waste (7% HsT + Come/Labo) counterparts. The greatest reductions in capillary porosity are found in the Come CSA cements, which exhibit a 2.32% decrease, and the Labo CSA cement, in which the figure falls by 1.05%.

With respect to macropore porosity (>50 nm), the mortars exhibit varying percentages. No clear trend is detected as regards the composition of the analyzed cements, with the hybrid mortars (7% HsT or HcG/Come CSA) having the highest porosity at 11.1% in both cases. However, the hybrid cements made with Labo CSA exhibit porosities similar to those of the other mortars. This has no correlation with the percentages of macropores in the pure Come CSA (6.15%) and Labo (26.4%) cements. However, the high percentage of macropores present in the CSA cement clinkerized at laboratory scale would explain the low mechanical compressive strength of this type of eco-cement.

The average pore sizes (4 V/A) run parallel to the macropore values to a certain extent: the greater the percentage of macropores, the greater the average pore diameter. Most of the analyzed mortars exhibit values around 0.05–0.06 µm. The only exception is the Labo CSA cement mortar, which presents a value of 0.32 µm consistent with the high porosity of this type of eco-cement.

The porosity of the cement and mortar is strongly related to mechanical properties. In the main, the relevant mechanical properties are influenced by the presence of macropores [48]. The results showed this behavior in the pure samples of OPC and CSA, where the mortars with lower amount of macropores, the Come CSA, presented a larger value of compressive strength and the mortar of Labo CSA, with a huge percentage of macropores, exposed a poor value of compressive strength.

Nevertheless, this behavior is altered when CDW is added to the mortar. The samples with 7% of HcG and HsT, with lower percentage of macropores, showed compressive strength values lower than OPC. This fact can be explained by the low reactivity of the CDW exhibit in the hydration heat, confirmed by the literature [15,16].

## 4. Conclusions

The following conclusions are drawn from the findings of this study:−Nearly all the ternary and hybrid cements present slightly lower water demand values than the OPC paste. The only exception is the Labo CSA cement, which requires 15.7% more water to achieve the same normal consistency. The use of CDW does not alter the initial setting times in any cements except the hybrid ones made with commercial CSA cement, which present reductions of around 130 min versus the OPC. The CDW materials used do not cause volume stability issues. All the ternary and hybrid cements meet the standardized physical requirements.−The capillary water absorption tests performed on the ternary and hybrid cements show very similar capillary behavior to the OPC during the first 40 min of the test. The CSA cement mortars and 7% HsT/HcG + 10% CSA (Labo y Come) hybrid cements had least capillary absorption capacity. The sorptivity coefficients obtained as a function of the fourth root show better R^2^ regression values than their square root counterparts.−The calorimetry tests performed using the standardized Langavant method reveal three different behaviors: (a) The ternary cement mortars (7% HsT and 7% HcG) behave similarly to the OPC; (b) the CSA cement mortars (Labo and Come) behave differently, with the commercial CSA cement (Come) achieving 370 J/g in the first 7 h, a value well above that obtained by the Labo CSA cement (90 J/g); and finally (c) the different hybrid cement mortars (OPC + CSA) exhibit two different calorimetric phases during the first 23 h that could be caused by the different hydration reactivities in the two cements.−The mechanical compressive strength values reveal that all the ternary and hybrid cement mortars containing CDW lost mechanical strength at both hydration ages versus the OPC mortar (35 and 73 MPa). At 28 d the values range from 58 to 65 MPa, with the values being slightly higher for the OPC + CSA cements. Despite this, almost all the analyzed mortars meet the mechanical requirements laid down in the European standard (≥52.5 MPa), the only exception being the Labo CSA cement mortar.−The total and partial porosity analyses are consistent with the mechanical results, revealing that the greatest differences are found in the macropores (>50 nm) and, in some cases, in the medium-size capillary pores. There is no clear trend with respect to the types of cements analyzed. However, the average pore diameter values are more uniform, ranging from 0.05 μm to 0.07 μm, except in the case of the laboratory-scale clinkerized CSA cement mortar, in which the value increases to 0.31 μm.

In light of the results obtained, it can be confirmed that the use of CDW, as a secondary raw material replacing natural raw materials in the manufacture of future ternary, CSA and hybrid eco-cements, is scientifically and technically feasible. This is a priority objective of environmental policy, the circular economy, and the Green Deal 2030 and Climate Neutrality 2050 strategies. It is, however, necessary to conduct further research into the scientific aspect of clinkerization of the new CSA eco-cement containing CDW to improve the ye’elemite/C_2_S ratio and the performance of the product with a view to its use in future construction applications.

## Figures and Tables

**Figure 1 materials-16-03093-f001:**
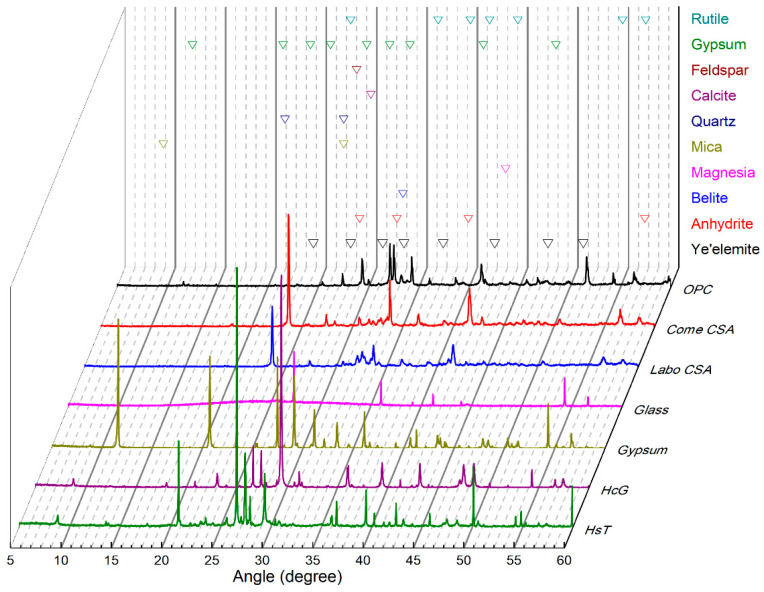
XRD analysis of raw materials.

**Figure 2 materials-16-03093-f002:**
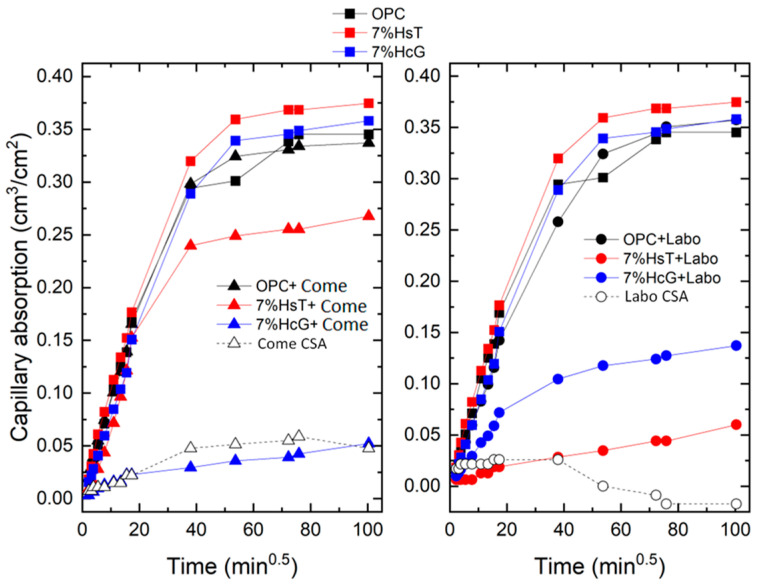
Capillary water absorption measurements for the mortars, employing the square root of the time.

**Figure 3 materials-16-03093-f003:**
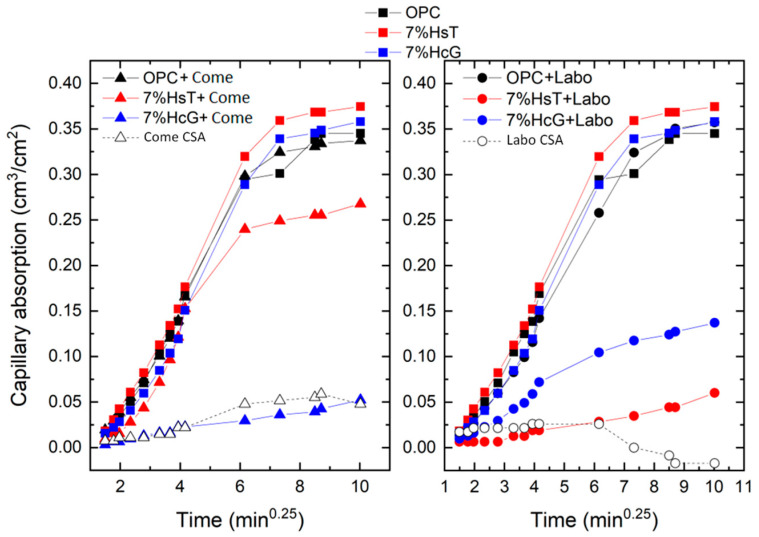
Capillary water absorption measurements for the mortars, employing the fourth root of the time [27].

**Figure 4 materials-16-03093-f004:**
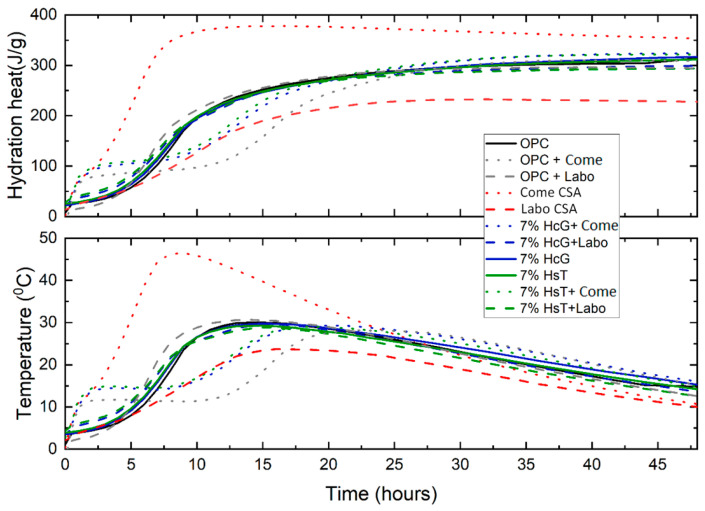
Hydration heat and temperature reached in the mortars, as measured by Langavant calorimetry.

**Figure 5 materials-16-03093-f005:**
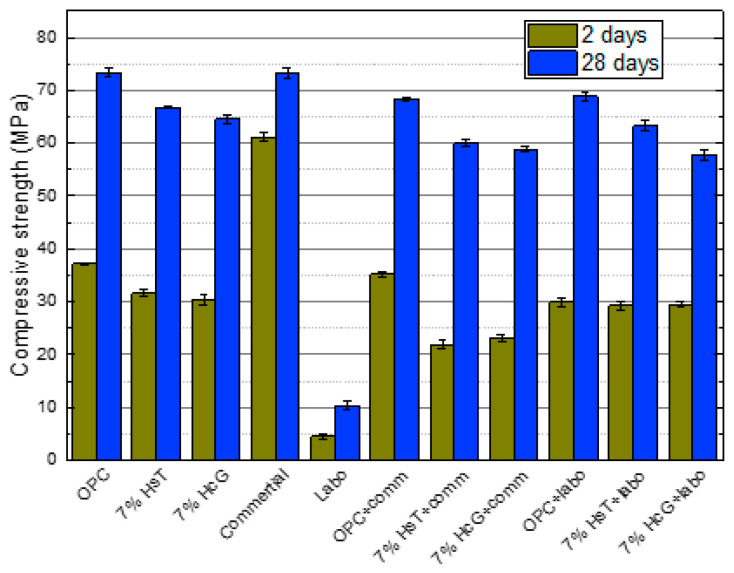
Compressive strength obtained at 2 and 28 days for all the eco-cements.

**Figure 6 materials-16-03093-f006:**
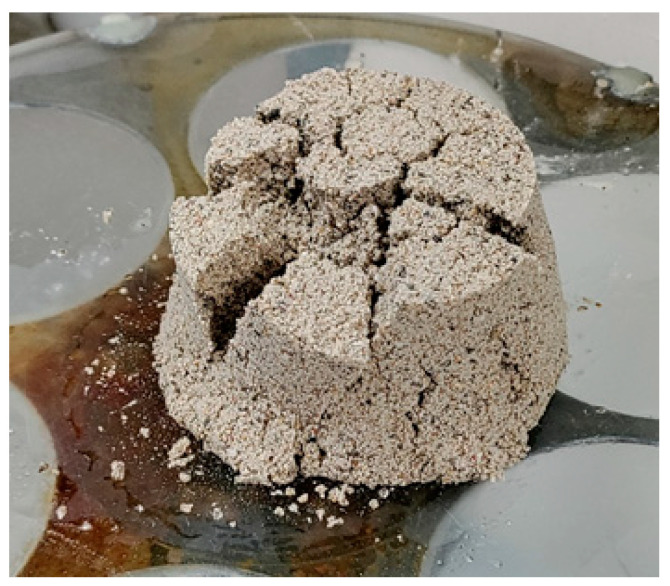
Appearance of the Labo CSA cement mortar on the shaker table.

**Figure 7 materials-16-03093-f007:**
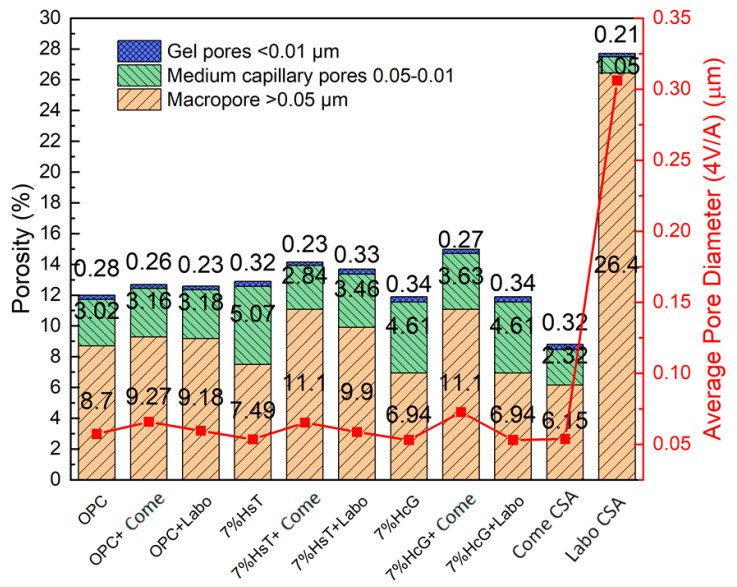
Porosity and average pore diameter at 28 days of curing.

**Table 1 materials-16-03093-t001:** Chemical composition of the starting materials, identified using XRF.

	Samples	CaO	Al_2_O_3_	SiO_2_	SO_3_	Fe_2_O_3_	MgO	TiO_2_	Na_2_O	K_2_O	LOI
	OPC	69.8	2.9	14.2	3.4	3.7	0.9	0.2	0.3	0.8	3.2
Cements	Come CSA	41.8	30.0	6.6	11.3	1.2	3.2	0.5	1.1	0.6	0.7
	Labo CSA	52.0	15.9	18.9	5.7	2.3	3.7	0.1	0.4	0.4	0.1
CDW	HsT	18.7	9.0	50.0	2.5	2.3	1.4	0.3	0.8	3.4	11.5
	HcG	50.3	2.9	9.3	0.9	1.2	1.1	0.1	0.2	0.5	33.2
	Gypsum	34.7	0.6	2.0	39.1	0.2	0.7	0.0	0.0	0.1	21.6
	Glass	9.6	1.1	70.3	0.2	0.9	3.6	0.1	13.3	0.3	0.4

**Table 2 materials-16-03093-t002:** Mineralogical composition of the CSA cements.

Cement/%	Ye’elemite	Anhydrite	C_2_S	Brediggite	Gehlenite
Labo CSA	23.58	2.38	15.94	27.85	30.24
Come CSA	92.39	4.44	3.17	0.00	0.00

**Table 3 materials-16-03093-t003:** D10, D50, and D90 values identified by laser granulometry.

µm	OPC	Come CSA	Labo CSA	Glass	Gypsum	HcG	HsT
Dx(10)	2.38	0.91	0.88	4.37	0.92	2.06	2.14
Dx(50)	12.50	4.47	5.04	24.3	3.68	11.9	12.4
Dx(90)	34.7	21.8	37.5	58.9	19.6	34.9	35.6

**Table 4 materials-16-03093-t004:** Dosages of the different cements.

Cements	OPC	HsT	HcG	Come CSA	Labo CSA	Glass
*OPC*	1	0	0	0	0	0
*7% HsT*	0.93	0.047	0	0	0	0.023
*7% HcG*	0.93	0	0.047	0	0	0.023
*OPC + 10% Come CSA*	0.90	0	0	0.10	0	0
*7% HsT + 10% Come CSA*	0.837	0.046	0	0.10	0	0.021
*7% HcG + 10% Come CSA*	0.837	0	0.046	0.10	0	0.021
*Come CSA*	0	0	0	1	0	0
*OPC + 10% Labo CSA*	0.90	0	0	0	0.10	0
*7% HsT + 10% Labo CSA*	0.837	0.046	0	0	0.10	0.021
*7% HcG + 10% Labo CSA*	0.837	0	0.046	0	0.10	0.021
*Labo CSA*	0	0	0	0	1	0

**Table 5 materials-16-03093-t005:** Water Demand (WD ± 1 g), Initial Setting Time (IST ± 10 min), and Soundness (S) of the cement pastes.

Cements	WD (g)	IST (min)	S (mm)
OPC	153	190	0.0
7% HsT	150	185	0.0
7% HcG	149	190	1.0
OPC + 10% Come	147	60	1.0
7% HsT + 10% Come	148	50	0.0
7% HcG + 10% Come	149	60	1.0
Come CSA	150	17	0.0
OPC + 10% Labo	147	180	1.0
7% HsT + 10% Labo	149	195	0.0
7% HcG + 10% Labo	149	185	0.0
Labo CSA	177	90	0.5
EN 197-1	-	≥45	≤10

**Table 6 materials-16-03093-t006:** Coefficient values corresponding to primary and secondary absorption sorptivity (AS).

	Primary AS (10^−2^ cm/min^0.5^)	R^2^	Secondary AS (10^−2^ cm/min^0.5^)	R^2^
OPC	0.966	0.996	0.095	0.812
7% HsT	0.129	0.996	0.080	0.734
7% HcG	0.843	0.990	0.099	0.745
OPC + Come	0.914	0.995	0.058	0.778
7% HsT + Come	0.910	0.977	0.042	0.984
7% HcG + Come	0.124	0.963	0.035	0.975
OPC + Labo	0.800	0.993	0.150	0.768
7% HsT + Labo	0.091	0.900	0.050	0.986
7% HcG + Labo	0.385	0.990	0.050	0.971
Come CSA	0.092	0.901	0.026	0.943
Labo CSA	0.046	0.737	−0.067	0.794

**Table 7 materials-16-03093-t007:** AS (×10^−2^ cm/min^0.25^), SD (×10^−2^), and R^2^ coefficient values according to Villagrán [27].

	OPC	7% HsT	7% HcG	OPC + Come	7% HsT + Come	7% HcG + Come	OPC + Labo	7% HsT + Labo	7% HcG + Labo	Come CSA	Labo CSA
AS	5.980	6.370	5.780	5.910	5.230	0.586	5.200	0.520	2.120	0.807	0.19
SD	0.220	0.270	0.380	0.270	0.320	0.048	0.260	0.054	0.095	0.101	0.043
R^2^	0.988	0.989	0.966	0.983	0.970	0.948	0.979	0.919	0.984	0.888	0.702

## Data Availability

Not applicable.

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
