# Peer review of "Physical and Mechanical Behavior of New Ternary and Hybrid Eco-Cements Made from Construction and Demolition Waste"

_materials, 2023, doi:10.3390/ma16083093_

Round 1

Reviewer 1 Report

Paper presents a topical issue. Article is written in clear language. The research design and methods are clearly stated. The arguments and discussion of findings are coherent, balanced and compelling. The conclusions are supported by the results presented in the article.  All the cited references are relevant to the research.

Remark:

Table 7 is not cited in the text.

Author Response

The authors would like to thank you very much for your comments on the research work presented.

Table 7 is not cited in the text. Included in the text in yellow color

Reviewer 2 Report

1.       Abstract 21-22. The results demonstrated within the abstract section sound too general, the Abstract section should include more precise results of the research.

2.       Introduction. It’s recommended to the authors to include more information re other researches connected with CO2 reduction by using eco-cements

3.       Methods does not include CO2 reduction estimation in case of CSA cement. However, the subject of the research is physical and Mechanical properties of the cements, but I would recommend adding the data about CO2 values of СSA cements to demonstrates the advantages for environment over ordinary cements. It may be added to the Introduction section or to the first part of Method section.

4.       Methodology section lacks more detailed description of the experimental methodology

Author Response

The authors would like to thank you very much for your comments on the research work presented. Modifications in the text are in blue colour

Comments and Suggestions for Authors

  1. Abstract 21-22. The results demonstrated within the abstract section sound too general, the Abstract section should include more precise results of the research.

It is difficult to introduce precise results when several different cements are analyzed. Nevertheless, some of the most relevant conclusions have been introduced in the Abstract section

  1. It’s recommended to the authors to include more information re other researches connected with CO2 reduction by using eco-cements

4 more references have been introduced in the Introduction section on the importance of active additions for CO2 reduction.

  1. Methods does not include CO2 reduction estimation in case of CSA cement. However, the subject of the research is physical and Mechanical properties of the cements, but I would recommend adding the data about CO2 values of СSA cements to demonstrates the advantages for environment over ordinary cements. It may be added to the Introduction section or to the first part of Method section.

We fully agree with your comment and request. The final objective of this research line is to obtain commercially viable eco-cements that meet the performance requirements of the standard and, of course, more circular and with less embodied carbon footprint.

We have prepared a specific paper focusing on the life cycle assessment of the CSA eco-cements (“NEW ECO-FRIENDLY CALCIUM SULFOALUMINATE CLINKERIZATION METHOD USING CONSTRUCTION AND DEMOLITION WASTE, AND PRODUCT LIFE CYCLE ASSESSMENT”). This paper concludes that the new eco-friendly CSA cement clinkers reduces the carbon footprint by 8.1% by using 20% of CDW for its clinkerization.

However, this paper is still under review so it can not be referred.

Concerning the binary cements blended with CDW, it was concluded in previous papers that the use of CDW as active addition reduces CO2 emissions in the same proportions as the percentage of clinker substitution. We have included some references at the introduction section.

  1. Methodology section lacks more detailed description of the experimental methodology

We agree that the methodology section can be improved in order to better understand the experimental methodology followed. We have tried to better organize the section and have add further specific information to better understand the experimental methodology. We consider that all the needed information is detailed in the section and previous works referenced. However, we remain open to incorporating specific additional information if it is still considered necessary.

Reviewer 3 Report

It’s honor to review this paper. In this article, mainly talking about the concrete mixed with Construction and Demolition Waste (CDW) and itself chemical, physical, and mineralogical characterization and make contrast with Ordinary Portland Cement and Calcium Sulfoaluminate Cement. The CDW concrete are more eco-friendly than the normal concrete. Therefore, it’s necessary to research the physical behavior, such as water demand, setting time, soundness, water absorption by capillary action, heat of hydration, and micro-porosity.

But overview this article, there is some improvement to apply in this article. The amendments are given below.

1.     Spelling error and words impropriety in this article;

2.     The expression is not clear in specific content at section 3, reader can’t to catch important message they need;

3.     The abbreviation ‘commercial CSA cement (Come CSA)’ at L343 not standard and easy to make misinterpretation, suggest to revise as ‘CCSA’ or others;

4.     It hard to repeat this experiment caused by the composition of Construction and Demolition Waste (CDW) are not embodied in article;

5.     In section 3.5, suggest to illustrate the influence of porosity on strength when using different type of concrete. And analysis the bond between concrete and CDW;

6.     Why the ability of capillary absorption shown as a linear in the first 6 hours, should to make an explanation;

7.     What condition the strength of the CDW concrete can get the perfect condition not show in the conclusion (section4), such as mix proportion, mixing amount of CDW in the concrete. And how many performances are promoted when adding the CDW into the concrete compare the normal commercial concrete, and the economic value are given by the CDW concrete.

Author Response

See attached letter
